# Determining the worldwide epidemiology of surgical site infections after gastrointestinal resection surgery: protocol for a multicentre, international, prospective cohort study (GlobalSurg 2)

GlobalSurg Collaborative

## ABSTRACT

**Introduction** Surgical site infection (SSI) is the most common complication following major gastrointestinal surgery, affecting between 25% and 40% of patients. The rate of SSI doubles from low-income to high-income settings, persisting after risk adjustment. The relative impact of antibiotic-resistant organisms and the effectiveness of antibiotic prophylaxis globally are unknown. This study aims to determine SSI rates following gastrointestinal surgery across worldwide hospital settings.

**Methods and analysis** This multicentre, international, prospective cohort study will be undertaken by any hospital providing emergency or elective gastroenterological surgical services. Centres will collect observational data on consecutive patients undergoing emergency or elective gastrointestinal resection, cholecystectomy or appendicectomy during a 6-month period. The primary outcome is the incidence of SSI with secondary outcomes describing the organisms causing SSIs, including their antibiotic susceptibility, and the microbiological tests used to identify them.

**Ethics and dissemination** This project will not affect clinical practice and has been classified as clinical audit following research ethics review. The protocol will be disseminated through the international GlobalSurg network.

**Trial registration number** NCT02662231.

### Strengths and limitations of this study

► This will be the first international, multicentre, prospective study to assess the incidence of surgical site infections (SSI) for patients undergoing elective and emergency major gastrointestinal surgery.

► The collaborative methodology adopted, as described elsewhere, allows for large volume, high-quality data collection while avoiding overburdening high-volume and low-resource centres that may otherwise be unable to participate in such projects.

► Previous studies have had varying subjective definitions of SSI; this study uses standardised definitions from the Centre for Disease Control and Prevention, which are widely implemented and internationally validated.

► Due to the observational nature of the study, it will not be possible to determine a causative link between risk factors and SSI and therefore comparisons of incidence and risk factors across different Human Development Index settings will be hypothesis generating only.

► As it is not possible to apply strict data monitoring within the confines of this study, we have developed a mixed-methods quantitative and qualitative validation process for our primary outcome measure, and case ascertainment rate, which will be applied to a sample of representative centres.

## INTRODUCTION

The burden of surgical disease in low and middle-income countries (LMICs) is growing.[1] Specific programmes have aimed to highlight the unmet need for safe surgery and anaesthesia as part of the global health agenda.[2] The Lancet Commission on Global Surgery outlined six core indicators for the assessment of global surgical systems, including the postoperative mortality rate (POMR).[3] While clearly important, mortality only affects 1%–4% of all patients.[4] For major gastrointestinal surgery, quantifying POMR alone neglects the burden of postoperative morbidity, which affects a greater proportion of patients.[2] Other relevant markers of postoperative outcome are also needed to determine the success of surgery in the majority of patients who will survive.

Surgical site infection (SSI) is the most common complication following major gastrointestinal surgery,[5] affecting between 25% and 40% of patients after midline laparotomy in high-income settings.[6 7] SSI is implicated in one-third of postoperative deaths and accounts for 8% of all deaths

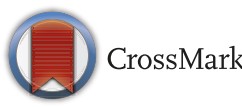

University of Birmingham, Birmingham, UK

**Correspondence to**
GlobalSurg Collaborative;
aneelbhangu@gmail.com

caused by a nosocomial infection.[8–10] Furthermore, SSIs cause pain and discomfort, increase hospital stay and put patients at greater risk of secondary infectious complications. This has an important economic impact with an attributable cost in the UK of £30 million per year.[11]

Our 2014/2015 prospective, observational cohort study (GlobalSurg 1[12]) included 10 475 patients from 58 countries. It showed that the incidence of SSI more than doubled from high (7.4%), to middle (14.4%), to low (20.0%)-income countries. This persisted after multivariable risk adjustment for patient and hospital confounders (middle income: OR 1.96 (1.63–2.32) and low income: OR 2.06 (1.67–2.57)). In the most contaminated procedures, one in three patients from LMICs suffered an SSI. Dirty surgery doubled in low-income countries (29.7% vs 16.6% in high-income settings), which was in turn associated with doubling of SSI (34.5% low income vs 15.4% high income). However, SSI was assessed as a secondary outcome measure as part of that study, lacking validity and requiring external validation.

Antibiotic-resistant organisms are now prevalent worldwide and a focus of interest for policy leaders and global health advocates.[13] In high-income settings, methicillin-resistant *Staphylococcus aureus* accounts for up to 50% of *S. aureus* infections.[14] While many hospitals in low-income settings are not able to routinely perform surveillance of resistant patterns, yet the little evidence that does exist suggests an accelerating incidence of antibiotic-resistant SSIs.[15] Patients with antibiotic-resistant infections have a higher risk of mortality and morbidity, and require more healthcare resources.[16] Currently, few data exist describing the international epidemiology of SSIs, the causative organisms or incidence of antimicrobial drug resistance. Investigating the diagnosis and treatment of SSIs is an urgent global health priority, with important implications for patients, clinicians and policymakers.

### Primary aims
The primary aim is to determine the incidence of SSI following elective and emergency gastrointestinal surgery across low, middle and high Human Development Index (HDI) countries.

### Secondary aims
The secondary aims include describing the organisms causing SSIs, the incidence of antibiotic-resistant pathogens and the microbiological tests used to identify them. The impact of the method of 30-day follow-up on these outcomes, where performed, will also be analysed.

### METHODS
### Study design
This is a multicentre, international, prospective, observational cohort study of all consecutive patients undergoing elective or emergency gastrointestinal surgery over a 14-day period. Individual collaborators are free to choose any 14-day period within the 6-month study period to collect data. The study will run from 4 January 2016 to 31 July (with 30-day follow-up of the last data collection period finishing on 30 August 2016). This 'snapshot' study design is a validated model that has been delivered successfully in previous studies.[6 17 18]

### The GlobalSurg network
GlobalSurg (http://globalsurg.org/) is an international research collaborative, fostering regional, national and international surgical networks. The collaborative model used has previously been described elsewhere[19] and has already facilitated one multicentre, international, prospective cohort study.[20] GlobalSurg is run by the Surgical Research Gateway (SuRG) Foundation (http://surg-foundation.org/; registered charity number 1159898), in collaboration with the University of Birmingham and University of Edinburgh. The objective of the charity is to advance the education of medical students and doctors in surgical science, clinical research and audit methods by promoting participation in collaborative clinical research and audit studies.

### Study setting
Any surgical unit providing emergency or elective surgery worldwide is eligible to participate. Included centres must capture all consecutive patients and ensure data collection is >95% complete. Centres with >5% missing data, when including all possible data points, will be excluded from the final analysis and removed from authorship. The minimum number of patients that must be recruited is 1. Multiple teams covering different non-overlapping time periods at each hospital are encouraged.

### Patient inclusion and exclusion criteria
Patients of all ages (adult and paediatric) undergoing elective, semielective or emergency gastrointestinal resection, cholecystectomy or appendicectomy are eligible to enter (box 1). Any operative approach can be used. Operations with a primary indication classified as vascular surgery, gynaecology, urology or transplantation should be excluded.

---

**Box 1  Patient inclusion and exclusion criteria**

Inclusion criteria
► Patients of all ages (adult and paediatric)
► Consecutive patients during a chosen 14-day study period
► Undergoing emergency or elective gastrointestinal resection, cholecystectomy and appendicectomy
► Includes open, laparoscopic, laparoscopic converted and robotic cases
► Primary indication of trauma should be included
► Hernia repair with bowel resection should be included

Exclusion criteria
► Operations with a primary indication classified as vascular surgery, gynaecology, urology or transplantation
► Caesarean sections
► Whipple procedure
► Simple hernia repair

## Box 2 Centre for Disease Control and Prevention (CDC) criteria for surgical site infection (SSI)[21]

CDC criteria for SSI require the patient to have at least one of the following:
► Purulent drainage from the superficial or deep (fascia or muscle) incision but not from within the organ/space component of the surgical site
► At least one of the following: pain or tenderness; localised swelling; redness; heat; fever and the incision is opened deliberately or spontaneously dehisces
► Abscess within the wound (clinical, histopathological or radiological detection)

Emergency procedures are defined as unplanned operations and include reoperations after previous procedures. Semielective procedures are defined as procedures planned on the background of the patient being discharged after an emergency admission in the past and is currently admitted for surgery. Gastrointestinal resection is defined as a complete transection and removal of a segment of the oesophagus, stomach, small bowel, colon or rectum.

Each individual patient should be included once into the study. Patients who return to theatre due to complications following earlier surgery can only be included if the index procedure is not already included.

### Outcome measures

The primary outcome measure is the incidence of superficial or deep incisional SSI within 30 days of surgery involving any of the operative incisions made. This measure adopts the definitions within the 2008 Centre for Disease Control and Prevention[21] definitions of SSI (box 2). Where it is unfeasible to follow-up the patients after discharge for 30 days postoperatively, in-hospital SSI incidence will be used as a proxy measure.

The secondary outcomes that will be derived from this study include:
1. a description of the global variation in organisms causing SSI (including use of microbiological tests);
2. the incidence of antibiotic-resistant pathogens detected in patients with SSI;
3. 30-day POMR;
4. 30-day postoperative reintervention rate;
5. assessing how difference in SSI rates varies based on the method by which patients are followed up.

### Data points

Data points relating to patient, immunosuppression status, operative method and postoperative period will be collected (table 1). In order to maximise data completeness, a minimal data set has been designed to test only factors relevant to the outcome measures. Investigators will enter data via the secure internet-based Research Electronic Data Capture (REDCap) system.[22] Anonymous patient data will be held on the system hosted by the University of Edinburgh, Scotland, UK.

### Investigators

The study will be undertaken by investigators around the world who will be responsible for disseminating the protocol at their individual site, ensuring appropriate study approvals are in place and collecting and uploading data to an online REDCap database. A central study writing committee compromising of an internationally representative group of healthcare professionals will be responsible for data analysis, final manuscript drafting and submission. Individuals will be required to register their unit via the REDCap system and will be required to complete two training modules prior to starting.

Countries with multiple sites will be assigned a country lead, who will be responsible for coordinating multiple teams across sites to ensure duplication of data does not occur. Each hospital will have a primary local investigator who will act as guarantor of data. A maximum of three local investigators can cover each 2-week data collection period. They will be responsible for gaining local audit, service evaluation or research ethics approval as appropriate to their institution. Investigators should also create clear mechanisms to identify and include all eligible patients, which involves daily review of operating logbooks, emergency admission lists and team handover lists. They should identify clear pathways to accurately collect baseline and follow-up data and be proactive in identifying postoperative events within the normal limits of follow-up.

Local arrangements for follow-up may include daily review of the patient and notes during admission and before discharge in order to identify in-hospital SSIs, reviewing patient status in outpatient clinics or via telephone interview at 30 days (if this normal practice), checking hospital notes (paper or electronic) and looking at handover lists. All investigators will be listed as collaborators on resulting publications.

### Quality of data

To ensure high data quality, a detailed protocol has been produced and published online. The protocol has been translated into common languages to ease investigator understanding, including Chinese (traditional and simplified), Spanish, Italian, Greek, Portuguese and Arabic.[23] In addition, investigators are mandated to complete two online educational modules, one a summary of the study protocol and the other an overview of the diagnosis and classification of SSI.

### Data validation

Data validation will be performed in two parts across a group of representative centres. Case completeness assessment will involve an independent investigator ascertaining the number of eligible cases within a 2-week data collection period at a participating centre and comparing this to actual number of cases submitted. Data accuracy will be assessed using qualitative, semistructured interviews with investigators to assess local data collection methods.

**Table 1** Required data fields

| Patient ID | Local hospital field |
| --- | --- |
| Age | If >2 years = whole years, if <2 years = months |
| Gender | Male, female |
| ASA score | I, II, III, IV, V, not recorded |
| Immunosuppression | Diabetes: Diet controlled, Tablet controlled, Insulin controlled, No<br>HIV: Yes—on antiretroviral therapy; Yes—not on antiretroviral therapy, No, Unknown<br>Steroids: Yes, No<br>Other immunosuppressive drugs (eg, azathioprine, methotrexate, biologic agents): Yes, No<br>Chemotherapy (current chemotherapy or if the last cycle was within 12 weeks of operation): Yes, No<br>Active malarial infection: Yes—confirmed by blood film or equivalent test, No |
|  | HIV: *Yes*—most recent preoperative CD4 count |
| Smoking status | Current smoker (including those who stopped smoking within the last 6 weeks), Previous smoker, Never smoked, Unknown |
| Date and time of admission | DD/MM/YYYY HH:MM |
| Date and time operation started (knife to skin time) | DD/MM/YYYY HH:MM |
| Length of operation (knife to skin until point of completion) | Minutes |
| Urgency of operation | Emergency (any surgery on the same admission as diagnosis), Semielective, Elective (any planned admission for surgery) |
| Was a surgical safety checklist used (WHO or an equivalent)? | Yes, No—but available in this centre, No—not available in this centre |
| Initial operative approach | Open midline, Open non-midline, Laparoscopic, Laparoscopic converted to open, Robotic, Robotic converted to open |
| Primary operation performed | Pick from drop-down list; pick single main procedure performed |
|  | Appendicectomy—Appearance at surgery: simple (non-perforated), complex (perforated, free pus), normal |
|  | Appendicectomy—Duration of symptoms (eg, abdominal pain) prior to surgery (days 0, 1, 2, 3, 4, 5, 6, 7+) |
| Main surgical pathology/indication (the main cause leading to surgery) | Malignant (proven or suspected tumour/cancer), Benign |
| Intraoperative contamination | Clean contaminated: GI tract entered but no gross contamination<br>Contaminated: GI tract entered with gross spillage or major break in sterile technique<br>Dirty: There is already contamination prior to operation (eg, with faeces or bile) |
| **Antibiotic use** |  |
| Used for treatment before surgery (eg, trial of antibiotics to treat diverticular abscess) | Yes (total days), No |
| Used for prophylaxis at the point of incision (ie, standard hospital prophylaxis) | Yes, No |
| Continued at the end of surgery (ie, extended prophylaxis after surgery) | Yes (total days), No |
| Was epidural analgesia inserted on the day of surgery? | Yes, No |
| Were NSAIDs used postoperatively during the first 5 days of after surgery? (including ibuprofen, naproxen, diclofenac, ketorolac and etoricoxib, excluding aspirin) | Yes, No |

Continued

**Table 1** Continued

| Patient ID | Local hospital field |
|---|---|
| Was serum haemoglobin/haematocrit checked in the first 48 hours postoperatively? | Yes—serum haemoglobin, Yes—capillary PCV, No—but tests available in this centre, No—tests not available in this centre |
| Was serum creatinine checked in the first 48 hours postoperatively? | Yes, No—but available in this centre, No—not available in this centre |
| Length of postoperative stay | Days |
| **Surgical site infection** | |
| Prior to discharge | Yes, No |
| At 30 days after surgery | Yes, No, Not assessed after discharge |
| | *If yes:* Was a wound swab sent for microbiological culture: Yes, No—but available in this centre, No—not available in this centre |
| | *If yes:* How was this treated: operative drainage, wound opened outside of operating theatre, antibiotics (tick all that apply) |
| What bacteria, if any, were identified? | None, *Staphylococcus aureus*, Coliform, Anaerobe, Other (five tick boxes) |
| | *If yes*: Sensitivity: sensitive to antibiotic prophylaxis given; resistant to antibiotic prophylaxis given; sensitivities not tested—but available in this centre; sensitivities not tested—not available in this centre |
| 30-day unexpected reintervention. Record the most serious reintervention | Yes—surgical, Yes—endoscopic, Yes—interventional radiology, No |
| 30-day mortality | Dead, Alive, Unknown. *If died:* postoperative day of death |
| 30-day intra-abdominal/pelvic abscess (CT, ultrasound or clinical (including reoperation) evidence of intra-abdominal or pelvic abscess) | Yes, No |
| Other hospital-acquired infection (treated with or without antibiotics) | Yes—urinary tract infection, Yes—pneumonia, Yes—central venous line infection, Yes—peripheral line infection, Yes—other, No |
| How was 30-day follow-up status achieved? (all applicable) | Still an inpatient, Clinic review, Telephone review, Community/home review, Discharged before 30 days and not contacted again |

ASA, American Society of Anaesthesiologists; GI, gastrointestinal; NSAID, non-steroidal anti-inflammatory drug; PCV, packed cell volume.

### Statistical analysis and power calculation

Identification of hospital or surgeon-specific performance will not be reported. Variation in outcome across different contexts will be tested using the HDI[24] (a composite statistic of life expectancy, education and income indices). Following analysis, centre-specific SSI rates will be fed back to participants. Hierarchical logistic regression multivariate analysis will be used to adjust the influence of HDI on SSI rates for confounding patient and disease variables. We will subgroup patients according to operative procedure for the purpose of analysis.

An appropriate sample size is difficult to calculate in this setting as there is a lack of previous study data to indicate the incidence of SSIs internationally. The previous GlobalSurg study recruited 10 745 patients using the same methodology.[25] Although the inclusion criterion for this study is different, we expect to achieve a similar number of patients. Based on the findings from GlobalSurg 1, this number will provide adequate power to show a minimum difference from 7% to 10% in SSI rate between two groups (80% power at 5% alpha requires a sample size of 1350 patients).

### Trial registration number

This study has been registered with ClinicalTrials.gov (identifier: NCT02662231). The registration is available to view via: https://clinicaltrials.gov/ct2/show/NCT02662231.

### Ethics and dissemination
#### Research ethics approval

The primary audit standard stems from the UK National Institute for Health and Clinical Excellence guidelines[26]

---

**Box 3  Study audit standards[27]**

Audit standards from National Institute for Health and Clinical Excellence: Surveillance
- ► People having surgery are cared for by healthcare providers that monitor surgical site infection rates (including postdischarge infections) and provide feedback to relevant staff and stakeholders for continuous improvement through adjustment of clinical practice (Quality Statement 7). Treatment of surgical site infection:
- ► People with a surgical site infection are offered treatment with an antibiotic that covers the likely causative organisms and is selected based on local resistance patterns and the results of microbiological tests (Quality Statement 6).

(box 3). As this study will not change local clinical practice and is limited to using data obtained as part of usual care, it has been classified as an audit by the South East Scotland Research Ethics Service in Edinburgh, Scotland (online supplementary file 1). Thus, this may be considered a global audit or global service evaluation.

Investigators are required to gain approval from the relevant responsible bodies, including local clinical audit departments, research and development departments or institutional review boards, as appropriate in their centre. If such departments are unavailable, written permission should be supplied by the chief of surgery or the responsible supervising consultant/attending physician.

### Protocol dissemination

The protocol will be disseminated across the established GlobalSurg network, compromised of surgeons, medical students and research nurses across the world. The network previously included over 1200 collaborators across 375 centres representing 58 countries.[12] Country leads are responsible for local coordination and dissemination within their country. In addition, the use of social media including Facebook, Twitter and YouTube has been shown to be an effective medium for dissemination of such collaborative projects[27] and will also be employed.

### Dissemination of results

We aim to publish the study results as open access. Data from the study will be described to ensure individual countries, hospitals and surgeons are anonymous and then shall be deposited in an online data repository for others to analyse. Based on the results of GlobalSurg 2 Study, a quality improvement protocol and/or interventional clinical trials will be suggested for possible application in the infection control units of each hospital included in this study.

### DISCUSSION

In this study protocol, we describe a multicentre, international, prospective cohort study investigating the incidence of SSI and its association with preoperative parameters and risk factors as well as with postoperative morbidity and mortality. Despite SSI being the most common complication following major gastrointestinal surgery, no data exist describing its global epidemiology. Similarly, appendicectomies and cholecystectomies are two of the most common intra-abdominal operations performed; yet the international burden of SSI for these patients remains unknown.

By using collaborative methodology[17] and a short 2-week data collection period, the study will recruit sufficient numbers of patients to inform this, while avoiding burdening low-resource centres that may otherwise be unable to participate. By investigating the morbidity and mortality caused by SSIs globally, this study will provide a platform to build future quality improvement programmes and interventional trials.

This study will be delivered using an international multidisciplinary collaborative network of healthcare researchers. The collaborative model has consistently proven potential for producing high-quality outcomes data with limited resource requirements across national[6 18 28] and international studies.[4] A detailed study protocol and mandatory completion of online training modules will ensure standardised definitions and understanding of SSI so that this study will deliver a reliable and accurate data set. A predefined data validation strategy will produce an estimate of case ascertainment and the quality of local methods used to gather data.

SSIs have been shown to increase costs by an average of US$20 842 in high-income countries, which significantly burden healthcare systems.[29] Within low-income countries, healthcare-associated infections have been estimated to have an increased incidence.[30] This can result in an increased length of stay of an average 9.7 days longer and has further knock-on effects to dependants which are more profound in the low-income setting.[29] To help address this fiscal disparity, this project will inform future service-level quality improvement programmes. When regular audits of pathogens and antibiotic prescribing are undertaken, redundant antibiotic use is reduced.[31 32] With feedback of baseline SSI rates to participating centres, collaborators have the opportunity to appraise their current practice against a global standard. Surgeons will be able to implement interventions to improve service provision prior to a reaudit, 1 year in the future. With the savings made from the implementation of prevention programmes being 11 times greater on average than costs,[33] these programmes will help reduce fiscal burden to patients and public health systems.

Many important determinants of a high burden of SSI in developing countries have not been identified.[5] This study will be able to identify an association between SSIs and various risk factors, however, due to its observational nature, it will not be able to prove causality. Therefore, data points collected have been included with a plausible relationship with SSIs. The number of data collection points has been kept short in order to maximise data completeness and accuracy and minimise investigator workload.

Finally, this study will continue to strengthen the GlobalSurg network, developing capacity for research in LMICs.[34] For the many research-naive centres worldwide, participation in this project could instil a culture of clinical effectiveness into routine clinical practice. This study will ultimately provide the basis for a global multicentre randomised controlled trial, aimed at reducing the burden of SSI and its attributed morbidity and mortality.

**Twitter** @GlobalSurg

**Acknowledgements** This project has received support from a DFID-Wellcome-MRC Joint Global Health Trials Development Award and a British Medical

Association grant. These organisations have not been involved in the drafting of this protocol or review of this manuscript.

**Collaborators** Chetan Khatri, Midhun Mohan, Thomas M Drake, James Glasbey, Dmitri Nepogodiev, Catherine Shaw, Zahra Jaffry, Stuart Fergusson, Francesco Pata, Adesoji O Ademuyiwa, Afnan Altamini, Hosni Khairy Salem, Andrew Kirby, Kjetil Soreide, Gustavo Recinos, Richard Spence, Sarah Rayne, Stephen Tabiri, Jen Cornick, Thomas Pinkney, Richard Lilford, J Edward Fitzgerald, Ewen M Harrison, Aneel Bhangu

**Contributors** All authors contributed to the design, drafting and review of this study protocol.

**Funding** This project has received support from the British Medical Association and the Association of Surgeons in Training. These organisations have not been involved in the drafting of this protocol or review of this manuscript.

**Competing interests** None declared. GlobalSurg is run by the Surgical Research Gateway (SuRG) Foundation. The SuRG Foundation is a registered charity (charity number 1159898) whose object is to advance the education of medical students and doctors in surgical science, clinical research and audit methods by promoting participation in collaborative clinical research and audit studies.

**Patient consent** This study was reviewed by the South East Scotland Research Ethics committee and they have classified this as clinical audit/service evaluation in the UK. As such, this does not require specific consent to be obtained from patients.

**Ethics approval** South East Scotland Research Ethics Service in Edinburgh.

**Provenance and peer review** Not commissioned; externally peer reviewed.

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
