## [Reviewer comments · BMJ Open]

ARTICLE DETAILS

TITLE (PROVISIONAL)	Determining the worldwide epidemiology of Surgical Site Infections after Gastrointestinal Resection Surgery: protocol for a multicentre, international, prospective cohort study (GlobalSurg 2).
AUTHORS	Khatri, Chetan; Collaborative, GlobalSurg; Bhangu, Aneel

VERSION 1 - REVIEW

REVIEWER	Daphne Roos Department of Surgery Reinier de Graaf gasthuis, Delft The netherlands
REVIEW RETURNED	03-Jul-2016

GENERAL COMMENTS	1. Why did the authors choose for a heterogenic population, including appendectomies and cholecystectomies and not just colorectal surgery? SSI after cholecystectomies and appendectomies are mostly diagnosed in outpatient settings, and may have no clinical importance, hence they will not prolong hospital stay. 2. As the authors already mentioned, the development of a SSI is a multimodal problem. Is it possible in this study to define all factors that play a role in the development of SSI? 3. Table 1: These factors are associated with an increase of SSI rate; -a high body mass index (BMI) - peri-operative hypothermia - excessive bloodless - low anastomosis in rectal surgery These factors should be included in table 1. 4. Perhaps include in table 1: The use of mechanical or antibiotic bowel preparation before surgery? Als Antibiotic use: Timing of prophylaxis antibiotics before incision needs to be noted (ideally 30-60 minutes before incision) also: type of antibiotics (is there aerobic and anaerobic coverage?) , single dose versus multiple dose. 5. Definition of anastomotic leakage differs in literature. What definition is used in this study?
---

REVIEWER	Virginia Shaffer Emory University USA
-----------------	---

REVIEW RETURNED	27-Aug-2016
-------------

GENERAL COMMENTS	Overall, the manuscript was well written. I had one question about the cases being grouped together. Will the cholecystectomy cases be grouped together with GI resection cases? These are very different cases and the GI resection cases would likely have more contamination than the cholecystectomy cases and thus different SSI rates. Will you be classifying cases- clean, clean-contaminated, or contaminated to stratify SSI rate?
--

VERSION 1 – AUTHOR RESPONSE

Reviewer 1:

1. Why did the authors choose for a heterogenic population, including appendectomies and cholecystectomies and not just colorectal surgery? SSI after cholecystectomies and appendectomies are mostly diagnosed in outpatient settings, and may have no clinical importance, hence they will not prolong hospital stay.

- We have chosen this specific study population to analyse their full impact at a worldwide level; this has not been performed in this setting. There is a lack of studies, which analyse the burden of SSI for appendectomies and cholecystectomies on an international setting. The capture of this data is a specific feedback point from our collaborators in LMICs. For readers to appreciate this, we have included this into our discussion.

We appreciate our study captures a mixed population group. We therefore intend to subgroup patients for the purpose of analysis and have clarified this in our methods.

2. As the authors already mentioned, the development of a SSI is a multimodal problem. Is it possible in this study to define all factors that play a role in the development of SSI?

- As mentioned in our strengths and limitations, due to observational nature of this study, it will not be possible to determine a causative link between risk factors and SSI. To clarify this, we have included this in our discussion.

3. Table 1:

These factors are associated with an increase of SSI rate;

- a high body mass index (BMI)
- peri-operative hypothermia
- excessive bloodless
- low anastomosis in rectal surgery

These factors should be included in table 1.

- We appreciate there are multiple factors that contribute to the development of SSI. It is not possible to capture every possible risk factor and we have chosen to keep our data points short in order to maximise data completeness whilst minimising investigator workload. Again, we have placed this into our discussion.

4. Perhaps include in table 1: The use of mechanical or antibiotic bowel preparation before surgery?

- Table 1 represents data points collected in the study. As the study has already begun, regrettably we are unable to add additional data points.

Als Antibiotic use: Timing of prophylaxis antibiotics before incision needs to be noted (ideally 30-60 minutes before incision) also: type of antibiotics (is there aerobic and anaerobic coverage?) , single dose versus multiple dose.

5. Definition of anastomotic leakage differs in literature. What definition is used in this study?

- Anastomotic leak is not a data point used in this study. We have therefore not provided a definition.

Reviewer 2:

Overall, the manuscript was well written. I had one question about the cases being grouped together. Will the cholecystectomy cases be grouped together with GI resection cases? These are very different cases and the GI resection cases would likely have more contamination than the cholecystectomy cases and thus different SSI rates. Will you be classifying cases- clean, clean-contaminated, or contaminated to stratify SSI rate?

- We appreciate our population is mixed and as detailed in our response above the burden of SSI for appendectomies and cholecystectomies has not yet been studied. We will subgroup these for the purpose of analysis and have clarified this in our methods.

We will stratify the analysis based on the intra-operative contamination.

VERSION 2 – REVIEW

REVIEWER	Daphne Roos Department of surgery Reinier de Graaf gasthuis city of Delft, The Netherlands
REVIEW RETURNED	29-Oct-2016

GENERAL COMMENTS	No other comments at this moment
----------------------------------

REVIEWER	Virginia Shaffer Emory University USA
REVIEW RETURNED	14-Nov-2016

GENERAL COMMENTS	Thank you for addressing previous concerns.
---